# A Phase II Study of S-1 plus Oxaliplatin for Patients with Recurrent Non-Squamous Cell Carcinoma of the Uterine Cervix (Tohoku Gynecologic Cancer Unit: TGCU206 Study)

**DOI:** 10.3390/cancers15215201

**Published:** 2023-10-29

**Authors:** Takayuki Nagasawa, Tadahiro Shoji, Eriko Takatori, Yoshitaka Kaido, Masahiro Kagabu, Dai Shimizu, Tatsuhiko Shigeto, Tsukasa Baba, Toru Sugiyama, Yoshihito Yokoyama

**Affiliations:** 1Department of Obstetrics and Gynecology, Iwate Medical University School of Medicine, Yahaba 028-3695, Japan; tnagasaw@iwate-med.ac.jp (T.N.); takatori@iwate-med.ac.jp (E.T.); ykaido@iwate-med.ac.jp (Y.K.); mkagabu@iwate-med.ac.jp (M.K.); babatsu@iwate-med.ac.jp (T.B.); 2Department of Obstetrics and Gynecology, Akita University School of Medicine, Akita 010-8543, Japan; shimizud1227@yahoo.co.jp; 3Department of Obstetrics and Gynecology, Hirosaki University School of Medicine, Aomori 036-8563, Japan; t-shigeto@hirosaki-u.ac.jp (T.S.); yokoyama@hirosaki-u.ac.jp (Y.Y.); 4Department of Obstetrics and Gynecology, St. Mary’s Hospital, Fukuoka 830-8543, Japan; sugiyamatoru0802@yahoo.co.jp

**Keywords:** cervical carcinoma, non-squamous cell carcinoma, recurrent, oxaliplatin, S-1, SOX

## Abstract

**Simple Summary:**

Cervical cancer is the fourth most common cancer in the world and the fourth leading cause of cancer death in women. In 2020, there were 604,000 new cases and 342,000 deaths from cervical cancer worldwide. Among these, non-squamous cell carcinoma accounts for about 20% of all cervical cancers and is increasing. It has long been known that non-squamous cervical cancer has a poorer prognosis than squamous cervical cancer. Reasons for this include relatively early lymph node metastasis and low sensitivity to radiotherapy. Therefore, chemotherapy is more likely to improve prognosis than radiotherapy. However, there are few studies on adenocarcinoma and a high level of evidence is not yet available. We have previously reported the efficacy of SOX (S-1+oxaliplatin) therapy for cervical adenocarcinoma in a pilot study. We report on a phase II trial of SOX therapy to evaluate its efficacy and safety.

**Abstract:**

Recurrent non-squamous cell carcinoma (non-SCC) of the uterine cervix is resistant to treatment and has a poor prognosis. The efficacy and safety of S-1/oxaliplatin (SOX) therapy in patients with recurrent non-SCC was examined in a phase II study. Fifteen patients were enrolled between August 2013 and March 2023. S-1 was administered orally at a daily dose of 80–120 mg for 14 days, and oxaliplatin was administered intravenously at a dose of 100 mg/m^2^ on day 1. Each treatment cycle lasted 21 days. The anti-tumor effects, adverse events, progression-free survival (PFS), and overall survival (OS) were investigated. The median patient age was 54 (41–74) years. The anti-tumor effect was rated as a partial response in five patients, stable disease in four, and progressive disease in 6. The overall response rate was 33% and the disease control rate was 60%. Regarding hematologic toxicities of grade 3 or more severity, leukopenia, neutropenia, anemia, and thrombocytopenia occurred in 26.6–40.0%. None of the patients discontinued the treatment because of adverse events. The median PFS and OS were 6 months (95% confidence interval [CI]: 2–11 months) and 22 months (95% CI: 11–23 months), respectively. No treatment-related deaths occurred. These results suggest that SOX therapy is useful for the treatment of recurrent non-SCC with promising anti-tumor effects and minimal adverse events.

## 1. Introduction

Cervical cancer is the fourth most common cancer worldwide and the fourth leading cause of cancer-related deaths due to cancer in women. In 2020, 604,000 new cases and 342,000 deaths due to cervical cancer were reported worldwide in 2020 [1]. In recent years, there have been an estimated 11,000 new diagnoses and 3000 deaths due to cervical cancer in Japan. Non-squamous cell carcinoma (SCC) accounts for 20.1% of all cervical cancers, and its incidence is increasing every year [2].

Non-SCC of the cervix has a poorer prognosis than SCC [3,4,5]. The reasons for this include relatively early lymph node metastasis [6] and low sensitivity to radiotherapy [7]. Therefore, chemotherapy, rather than radiation or chemoradiation therapy, is expected to improve prognosis. New treatment strategies need to be attempted; however, there are few studies on adenocarcinoma, and a high level of evidence is not yet available. We have previously reported the efficacy of S-1 plus oxaliplatin (SOX) therapy for cervical adenocarcinoma in a pilot study [8]. Here, we report a phase II trial of SOX therapy at participating TGCUs to evaluate its efficacy and safety.

## 2. Materials and Methods

### 2.1. Study Population

A multicenter clinical study was conducted among patients with recurrent cervical non-SCC who were enrolled in the study between August 2013 and March 2023 and met the following criteria: (1) non-SCC confirmed by histological diagnosis (neuroendocrine tumors were excluded), (2) recurrence after previous chemotherapy, (3) measurable or evaluable lesion, (4) European Oncology Cooperative Group performance status 0–2, (5) age between 20 and 75 years, (6) expected survival time of >3 months, (7) no effect on major organ functions (white blood cell count ≥ 3000/mm^3^, neutrophil count ≥ 1500/mm^3^, platelet count ≥ 100,000/mm^3^, hemoglobin ≥ 9.0 g/dL, total bilirubin ≤ 1.5 mg/dL, serum creatinine ≤ 1.5 mg/dL, creatinine clearance ≥ 60 mL/min, and (8) informed consent provided. The exclusion criteria were: (1) Grade 2 or higher peripheral neuropathy, (2) evident pulmonary fibrosis or interstitial pneumonitis, (3) pleural or cardiac effusion necessitating prompt local treatment, (4) brain metastasis requiring prompt treatment, (5) diarrhea (watery stool), (6) intestinal paralysis or intestinal obstruction, (7) active infection requiring treatment with antimicrobial agents, and (8) patients considered inappropriate as subjects by the physician in charge for any other reason.

### 2.2. Treatment

S-1 was orally administered twice daily for 14 consecutive days (from the evening of day 1 to the morning of day 15). Oxaliplatin was administered on day 1. The dose regimen of S-1 was 80–120 mg/day (40–60 mg per dose). More specifically, the S-1 dose was 80 mg/body/day for patients whose body surface area (BSA) was less than 1.25 m^2^, 100 mg/body/day for those who had a BSA of 1.25 to less than 1.5 m^2^, and 120 mg/body/day for those with a BSA of 1.5 m^2^ or more. Oxaliplatin, 100 mg/m^2^ dissolved in 250 mL of 5% glucose, was administered intravenously for 120 min. Each cycle of chemotherapy lasted 21 days, and treatment was continued until the occurrence of serious adverse events (AEs), making it difficult to continue treatment, or until disease progression.

The requirements for therapy initiation were as follows: neutrophil count, 1500/mm^3^ or more; platelet count, 75,000/mm^3^ or more; AST/ALT, less than 100 IU/L (less than 150 IU/L if liver metastases present); total bilirubin, less than 1.5 mg/dL; serum creatinine, less than 1.5 mg/dL; and no signs of diarrhea or infection. The requirements for the initiation of the next cycle were as follows: neutrophil count, 1500/mm^3^ or more; platelet count, 75,000/mm^3^ or more; and absence of non-hematologic toxicities of grade 2 or more, excluding nausea, vomiting, anorexia, fatigue, and hair loss.

In patients with grade 4 thrombocytopenia in the previous cycle or whose platelet count did not reach 75,000/mm^3^ on the day scheduled for initiation of the next cycle but who showed a restored count within 14 days, the dose of oxaliplatin was reduced from 100 mg/m^2^ to 75 mg/m^2^. When the platelet count did not reach 75,000/mm^3^ even after 14 days, the therapy was discontinued. When grade 4 neutropenia persisted for at least 5 days or when febrile neutropenia of grade 3 or higher severity occurred, the dose of oxaliplatin was reduced from 100 mg/m^2^ to 75 mg/m^2^, and the dose of S-1 was reduced by one level (e.g., a preceding dose of 120 mg/body/day was decreased to 100 mg/body/day, and likewise 100 mg/body/day to 80 mg/body/day and 80 mg/body/day to 50 mg/body/day). When a grade 2 sensory nerve disorder occurred and did not improve to grade 1 by the day scheduled for the initiation of the next cycle, the oxaliplatin dose was reduced from 100 mg/m^2^ to 75 mg/m^2^. When the same symptoms developed a second time, the oxaliplatin dose was reduced from 75 mg/m^2^ to 50 mg/m^2^. If the same symptoms were observed a third time, therapy was discontinued. When a grade 3 sensory nerve disorder occurred but had improved to grade 1 by the day scheduled for the initiation of the next cycle, the oxaliplatin dose was reduced as described above. If the symptoms did not improve to grade 1, therapy was discontinued.

### 2.3. Sample Size

The response rate to single-agent therapy with second-line chemotherapy for cervical cancer is approximately 15% [9,10,11]. In the JCOG0505 study of advanced and recurrent cervical cancer in Japan, the response rates to TP and TC therapy were 58.8 and 62.8%, respectively [12]. However, considering that patients with a history of chemotherapy had a lower response rate, the threshold response rate was set at 15%: The expected response rate was 30%, α (one-sided) = 0.05, and the power at 80%. Using the MinMax two-stage design of Simon, the required number of patients was calculated to be 48. Considering the ineligible and excluded patients, the final target number of patients was set at 52.

### 2.4. Endpoints/Variables

The primary endpoint was progression-free survival (PFS), and the secondary endpoints were anti-tumor efficacy, AEs, and overall survival (OS). The anti-tumor effects were evaluated using the Response Evaluation Criteria in Solid Tumor (RECIST) version 1.1 [13]. The adverse events were graded according to the Japanese version of the National Cancer Institute Common Toxicity Criteria (NCI-CTCAE) version 5.0 JCOG Japanese version (CTCAE v5.0-JCOG) [14]. PFS and OS were calculated using the Kaplan–Meier method.

### 2.5. Statistical Analysis

The data cut-off date was 31 March 2023. PFS and OS were calculated from the start of chemotherapy to the documented date of progression, death, or last follow-up, whichever occurred first. The impact of chemotherapy on survival was assessed by constructing Kaplan–Meier curves using a log-rank test. All statistical analyses were performed using EZR (Saitama Medical Center, Jichi Medical University, Saitama, Japan), a graphical user interface for R (The R Foundation for Statistical Computing, Vienna, Austria). More precisely, it is a modified version of the R commander, designed to add statistical functions frequently used in biostatistics [15].

## 3. Results

### 3.1. Patient Characteristics

Fifteen patients were enrolled, and none were ineligible. The median age of the patients was 54 years (range, 41–74 years), and the PS was 0 in 11 patients and 1 in 4 patients. Five patients had clinical stage I disease, 3 had stage II disease, 1 had stage III disease, and 6 had stage IV disease. The histological type was adenocarcinoma in 12 patients and clear cell carcinoma, endometrioid carcinoma, and unclassified in 1 patient. One case of “unclassified” cancer was a stage IVB case. A cervical biopsy was performed, and immunostaining was positive for CAM5.2 and negative for 34bE12, p63, CK14, and HIK1083. Squamous cell carcinoma was ruled out, although further investigation was not possible because of low specimen volume. After initial chemotherapy, the uterus was removed to control bleeding; however, owing to chemotherapy, the adenocarcinoma subtype could not be diagnosed. The number of previous therapeutic regimens was less than three in nine patients and three or more in six, and the platinum-free interval was less than six months in eight and six months or more in seven patients. Ten patients had undergone previous radiotherapy, while five patients had not. Local recurrence and distant metastases were observed in ten patients (Table 1).

### 3.2. Survival Analysis

The median follow-up period was 14 months (range, 2–23 months), and the median PFS and OS were 6 months (95% confidence interval [CI]: 2–11 months) and 22 months (95% CI: 11–23 months), respectively (Figure 1).

### 3.3. Toxicity

Regarding hematologic toxicities of grade 3 or higher severity, leukopenia occurred in 4 (26.6%) and neutropenia in 6 (40.0%) patients. In one case of grade 3 febrile neutropenia, the dose of S-1 was reduced by one level from the next cycle. In one case of grade 4 neutropenia lasting >5 days, the dose of oxaliplatin and S-1 was reduced by one level from the next cycle. Grade 3 or more severe developed in 6 (40.0%). In these patients, the platelet count was restored to more than 75,000/mm^3^ on day 28, without platelet transfusion. However, three patients received one level of oxaliplatin from the next cycle and did not experience grade 4 thrombocytopenia thereafter. The other three patients were diagnosed with progressive disease (PD) and treatment was terminated. Grade 3 or more severe anemia occurred in six (40.0%) patients, and all patients received blood transfusions. One of these patients received two cycles of transfusion.

Regarding non-hematologic toxicities, grade 4 cerebral infarction occurred in one patient (6.7%) who discontinued treatment after 2 cycles. The other non-hematologic toxicity of grade 3 or higher was nausea and fatigue, which occurred in one patient (6.7%). In this patient, oxaliplatin was discontinued after the 9th cycle and S-1 was continued until the 12th cycle. Grade 1 and 2 sensory neuropathy developed in all patients (100%) (Table 2).

A delay in the next cycle was observed in six patients. One of these patients was deferred after two cycles because the neutrophil count was <1500/mm^3^ and the platelet count was <75,000/mm^3^. Four patients had only one cycle delay because three had neutrophil counts of less than 1500 and one had neutrophil counts of less than 1500/mm^3^ and platelet counts of less than 75,000/mm^3^. These patients started the next cycle after 7 days. The remaining patient had lower gastrointestinal bleeding and chemotherapy was delayed because of hemostatic treatment. No treatment-related deaths occurred.

### 3.4. Anti-Tumor Response

The median number of cycles of SOX therapy was five (range, 2–12). The anti-tumor effect of RECIST version 1.1 was evaluated as partial response (PR) in 5 patients, stable disease (SD) in 4 patients, and progressive disease (PD) in 6 patients. The overall response rate was 33% (95% CI: 38.4–88.2), and the disease control rate was 60% (95% CI: 16.3–67.7) (Table 3).

## 4. Discussion

Previously, chemotherapy for non-SCC of the cervix was administered as a single agent or as a cisplatin-based combination regimen [16,17,18,19,20]. Recently, the efficacy of chemotherapy using a combination of taxanes and platinum agents has been reported [21,22,23,24]. The GOG240 study was a randomized control trial of 452 patients with advanced cervical cancer, including 86 patients with adenocarcinoma, which demonstrated that chemotherapy with bevacizumab improved OS (16.8 months in the chemotherapy plus bevacizumab groups versus 13.3 months in the chemotherapy-alone groups (hazard ratio 0.77 (95% CI 0.62–0.95); *p* = 0.007) [25].

However, there is no established second-line chemotherapy for cervical non-SCC that recurs after primary treatment with taxanes and platinum. Takatori et al. [8] conducted a pilot study of SOX therapy for recurrent cervical adenocarcinoma and suggested that SOX therapy may be useful. We conducted a prospective TGCU study to investigate the efficacy and safety of SOX therapy. The planned enrollment was 52 patients; however, owing to low enrollment and financial problems, the current study was terminated after 15 patients were enrolled. No cases of CR were observed in this study; however, five cases of PR and four cases of SD were observed. The response and disease control rates were 33% and 60%, respectively, indicating a relatively good anti-tumor efficacy in patients with relapsed disease. Of the six (40%) patients with grade 3 or more severe neutropenia, two patients had a one-step dose reduction of S-1 alone or S-1 and oxaliplatin. Severe thrombocytopenia of grade 3 or higher occurred in six patients (40.0%). In these patients, platelet counts recovered to >75,000/mm^3^ on day 28, without platelet transfusion. However, three patients had their oxaliplatin dose reduced in one step, beginning with the next cycle. Grade 3 or more severe anemia was observed in six patients (40.0%), and all patients received blood transfusions. One possible reason for such a large number of hematologic toxicities is the large number of patients with a history of radiotherapy (10 patients) and the inclusion of patients who had received more than three regimens of chemotherapy. In fact, only one patient with no history of radiation therapy and fewer than two chemotherapy regimens developed grade 3 or higher hematologic toxicity. Although the rates of grade 3 severe anemia and thrombocytopenia were higher than those reported in the pilot study by Takatori et al. [8], it is difficult to make a comparison because the history of prior radiation therapy was not known in the pilot study report, and the delimitation of the number of prior regimens was different. With regard to non-hematologic toxicity, grade 1–2 sensory neuropathy specific to oxaliplatin therapy occurred in all patients; however, no patient discontinued treatment because of this. There were no treatment discontinuations or deaths due to hematologic or non-hematologic toxicities.

Recently, anti-PD-1 antibodies have been used to treat cervical cancer. The EMPOWER-Cervical 1/GOG-3016/ENGOT-cx9 study was a randomized trial of 608 patients with recurrent cervical cancer that had progressed on platinum-based therapy (77.8% patients had squamous cell carcinoma and 22.2% patients had adenocarcinoma), comparing cemiplimab to single-agent chemotherapy (physician’s choice), demonstrated a significant increase in overall survival (12.0 months for cemiplimab vs. 8.5 months for single-agent chemotherapy). In particular, overall survival was longer in patients with adenocarcinoma by histologic subtype (13.3 months for cemiplimab vs. 7.0 months for single-agent chemotherapy) [26]. Therefore, cemiplimab is a promising second-line or later regimen for recurrent cervical cancer, but the response rate is 16.4%, which is not high. These costs must also be considered. As it is a newly approved drug, it is expensive and requires approximately 450,000–500,000 yen per cycle in Japan. SOX therapy is a combination of the conventionally used oxaliplatin infusion and oral S-1, and the cost of one cycle is generally less than JPY 100,000. Although the number of patients in this study was small and the reliability was not sufficient, SOX therapy had a response rate of 33% and a disease control rate of 60%. Considering its anti-tumor effect and cost-effectiveness, it may be considered a treatment option for recurrent cervical cancer.

Pembrolizumab is another anti-PD-1 antibody used to treat cervical cancer. The KEYNOTE-826 study reported that pembrolizumab plus platinum-based chemotherapy as a first-line treatment for cervical cancer with residual, recurrent, or metastatic disease significantly extended OS and PFS compared to those without pembrolizumab [27]. This suggests that the use of anti-PD-1 antibodies in first-line therapy may increase in the future. However, the efficacy of cemiplimab, an anti-PD-1 antibody, in relapse after anti-PD-1 antibody administration has not been clarified. This is one of the reasons why SOX therapy is favored.

Here, we discuss this from a different perspective. Prevention is essential in the initial management of cancer. Among gynecological cancers, accurate management of cervical cancer is particularly important. Primary and secondary prevention of cervical cancer includes vaccination and screening tests, such as cytology and human papillomavirus (HPV) testing. The strict implementation of these preventive measures can reduce the incidence and mortality rates of cervical cancer. However, this prevention strategy is inconsistent worldwide and tends to be particularly inadequate in low-income countries [28]. Furthermore, there are even high-income countries such as Japan, where few people receive vaccinations or screening tests. The incidence of advanced cervical cancer has not decreased in these regions. Therefore, it is necessary to consider appropriate treatment methods for patients with advanced and recurrent cervical cancer. The presence of HPV, various genetic abnormalities, and immune responses associated with cervical cancer have been reported. It may be desirable to strive to generate individualized medical care for each patient according to their status. It is important to continue research to perfect the tailored treatment for each patient based on their characteristics. However, given the large number of patients in low-income countries, treatments that are easy to manage and inexpensive are undeniably needed. This need may also be recognized in high-income countries.

This study has several limitations. First, quality of life (QOL) evaluation is important in the treatment of recurrent disease, and it is regrettable that we did not perform a QOL evaluation. However, because the intravenous administration time of oxaliplatin is approximately 120 min including premedication, outpatient treatment is possible, and serious adverse events with SOX therapy are rare and manageable. Thus, it is assumed that the patients were able to maintain a high QOL. Second, the most serious limitation is that the present study was designed as a prospective study but was terminated owing to lower-than-expected enrollment and financial concerns. Although a sufficiently long enrollment period was provided, patient enrollment did not increase and the decision to discontinue enrollment was inevitable. Based on the study design, we planned to enroll 48–52 participants, considering a priori assumption and power analysis; therefore, the enrollment of 15 participants was insufficient for statistical validation. Therefore, the response and disease control rates of SOX therapy in the present study were 33% and 60%, respectively, which seems to be a relatively good anti-tumor effect for recurrent non-SCC, but these results remain statistically uncertain. However, the data are interesting given the paucity of active regimens against recurrent cervical adenocarcinoma. We understand the opinion that results that cannot be statistically confirmed should not be published. However, leaving the trial data unpublished was contrary to the beliefs of the researchers and participants. Additionally, “Ethical Concerns if Clinical Trial Results Go Unreported” mentions the comment of Harlan Krumholz, MD, Director of the Center for Outcomes Research and Evaluation at Yale. He said, “The failure to report trial results is the betrayal of the participants and a skewing of the evidence that the public can see” [29]. Furthermore, the Declaration of Helsinki, adopted by the World Medical Association in 1964 and amended in 2000, states that the ethical imperative to report includes the results of unreported trials conducted in the past. Negative, inconclusive, or positive results must be published [30]. Referring to these statements, we decided to report the results of the present study with a belief in the potential of SOX therapy.

## 5. Conclusions

SOX therapy for non-SCC of the cervix may contribute to the treatment strategy in terms of anti-tumor effect and cost-effectiveness, and its adverse effects are manageable. Therefore, SOX therapy may serve as a new treatment option.

## Figures and Tables

**Figure 1 cancers-15-05201-f001:**
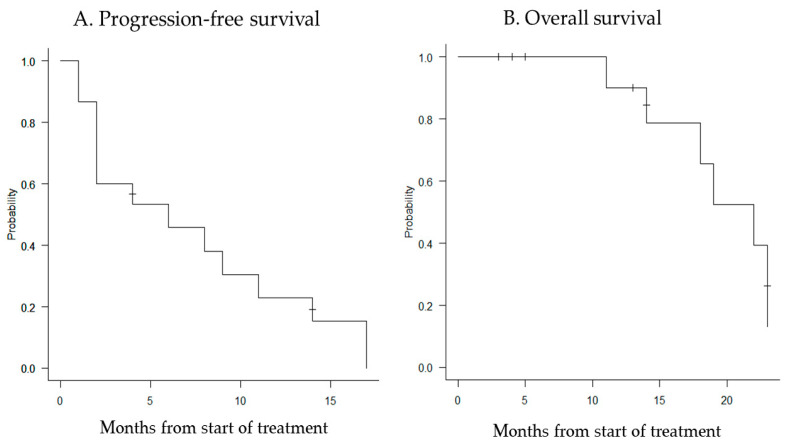
Kaplan–Meier curves for progression-free survival (**A**) and overall survival (**B**). The median PFS for all patients was 6 months (95% CI: 2–11 months), and the median OS was 22 months (95% CI: 11–23 months).

**Table 1 cancers-15-05201-t001:** Patient characteristics.

Age	Median	54
	Range	41–74
PS	0	11
	1	4
stage	I	5
	II	3
	III	1
	IV	6
Histological type	Adenocarcinoma	
	usual type	12
	Clear cell	1
	Endometrioid	1
	Unclassified	1
Number of prior regimens	<3	9
	≥3	6
History of radiotherapy	Yes	10
	No	5
Platinum-free interval	<6	8
	≥6	7
Recurrence site	Local	10
	Distant	10

**Table 2 cancers-15-05201-t002:** Toxicity.

	Grade 1	Grade 2	Grade 3	Grade 4	≥Grade 3 (%)
Hematologic toxicity					
Leucopenia	4	6	3	1	4 (26.7)
Neutropenia	3	2	4	2	6 (40.0)
Febrile neutropenia	0	0	1	0	1 (6.7)
Anemia	3	5	3	3	6 (40.0)
Thrombocytopenia	3	3	1	5	6 (40.0)
Non-hematologic Toxicity					
Nausea	6	1	1	0	1 (6.7)
Vomiting	0	0	0	0	0 (0)
Diarrhea	3	3	0	0	0 (0)
Sensory neuropathy	12	3	0	0	0 (0)
Mucositis	5	0	0	0	0 (0)
Appetite loss	14	0	0	0	0 (0)
Fatigue	8	5	1	0	1 (6.7)
Cerebral infarction	0	0	0	1	1 (6.7)

**Table 3 cancers-15-05201-t003:** Anti-tumor Response.

CR (N = 0)	0%
PR (N = 5)	33%
SD (N = 4)	27%
PD (N = 6)	40%
ORR (N = 5)	33%
DCR (N = 9)	60%

CR, complete response; PR, partial response; SD, stable disease; PD, progressive disease. ORR objective response rate, DCR disease control rate.

## Data Availability

The data presented in this study are available on request from the corresponding author. The data are not publicly available due to the protection of patient identification.

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
