# Peer review of "A Phase II Study of S-1 plus Oxaliplatin for Patients with Recurrent Non-Squamous Cell Carcinoma of the Uterine Cervix (Tohoku Gynecologic Cancer Unit: TGCU206 Study)"

_cancers, 2023, doi:10.3390/cancers15215201_

Round 1
Reviewer 1 Report
Comments and Suggestions for Authors
Very interesting article
Reviewer 2 Report
Comments and Suggestions for Authors
I read with great interest the Manuscript titled " A phase II study of S-1 plus oxaliplatin for patients with recurrent non-squamous cell carcinoma of the uterine cervix 3 (Tohoku Gynecologic Cancer Unit: TGCU206 study)." topic interesting enough to attract readers' attention.
Although the manuscript can be considered already of good quality, I would suggest to take into account the following recommendations:
- I suggest round of language revision, in order to correct few typos and improve readability;
- I suggest that authors to add a reference to highlight the importance of precise management in cervical cancer patients and, considering results of this study, to discuss how each patient's tailored-treatment could be perfected based on their characteristics and recurrences pattern. I would be glad if the authors discuss this important point, referring to: Tullio Golia D'Augè, Andrea Giannini, Giorgio Bogani, Camilla Di Dio, Antonio Simone Laganà, Violante Di Donato, Maria Giovanna Salerno, Donatella Caserta, Vito Chiantera, Enrico Vizza, Ludovico Muzii, Ottavia D’Oria. Prevention, Screening, Treatment and Follow-Up of Gynecological Cancers: State of Art and Future Perspectives. Clin. Exp. Obstet. Gynecol. 2023, 50(8), 160. https://doi.org/10.31083/j.ceog5008160.
Because of these reasons, the article should be revised and completed. Considering all these points, I think it could be of interest to the readers and, in my opinion, it deserves the priority to be published after minor revisions.
Comments on the Quality of English LanguageI suggest round of language revision, in order to correct few typos and improve readability
Reviewer 3 Report
Comments and Suggestions for Authors
The authors present the results of their phase II study of S-1 plus oxaliplatin for patients with recurrent non-squamous cell carcinoma of the uterine cervix.
Overall, the study was classically designed and the manuscript is well-written. With regard to the results, my only question concerns the histology; why was a patient with "unknown" histology included?
As the authors comment, the primary issue with this study is the severe lack of subjects. According to the study design, 48-52 subjects were planned given the a priori assumptions and power analysis. However, "owing to low enrollment and financial problems, the current study was terminated after 15 patients were enrolled." Clearly, this is a serious issue.
On one hand, the study as reported is severely underpowered and the conclusions unsupported. While the observed "response and disease control rates were 33% and 60%", these rates are not statistically supported. Thus, the authors' conclusion that these rates indicate "a relatively good anti-tumor efficacy in patients with relapsed disease" is not statistically valid.
On the other hand, the data are interesting, especially given the paucity of active regimens against recurrent adenocarcinoma of the cervix. In addition, "Study participants believe investigators are conducting their research to promote the public good and scientific advancement. But leaving trial data unpublished creates its own kind of bias, possibly harming the public.[https://www.reliasmedia.com/articles/147474-ethical-concerns-if-clinical-trial-results-go-unreported, https://www.alltrials.net/find-out-more/why-this-matters/obligations-to-report/]
In this case, I will have to defer to the editors to decide if this study is published. My belief is that it should be published, but if so, the authors MUST thoroughly describe the impact of the inadequate numbers.
Round 2
Reviewer 3 Report
Comments and Suggestions for Authors
The authors have adequately addressed my concerns. I support publishing the manuscript. If so, moderate editing for English language and grammar is necessary, especially the updated sections.
Comments on the Quality of English LanguageModerate editing for English language and grammar is necessary, especially the updated sections.
